# PLEX: Making the Most of the Available Data for Robotic Manipulation Pretraining

**Garrett Thomas**[*]     **Ching-An Cheng**[†]     **Ricky Loynd**[†]     **Felipe Vieira Frujeri**[†]

**Vibhav Vineet**[†]     **Mihai Jalobeanu**[‡]     **Andrey Kolobov**[†]

**Abstract:** A rich representation is key to general robotic manipulation, but existing approaches to representation learning require large amounts of multimodal demonstrations. In this work we propose PLEX, a transformer-based architecture that learns from a small amount of *task-agnostic visuomotor* trajectories and a much larger amount of *task-conditioned* object manipulation *videos* — a type of data available in quantity. PLEX uses visuomotor trajectories to induce a latent feature space and to learn task-agnostic manipulation routines, while diverse video-only demonstrations teach PLEX how to plan in the induced latent feature space for a wide variety of tasks. Experiments showcase PLEX's generalization on Meta-World and SOTA performance in challenging Robosuite environments. In particular, using relative positional encoding in PLEX's transformers greatly helps in low-data regimes of learning from human-collected demonstrations.

**Keywords:** Robot learning, Robotic manipulation, Visuomotor representations

## 1 Introduction

Transformers [1] have lead to breakthroughs in training large-scale general representations for computer vision (CV) and natural language processing (NLP) [2], enabling zero-shot adaptation and fast finetuning [3]. At the same time, despite impressive progress, transformer-based representations haven't shown the same versatility for robotic manipulation. Some attribute this gap to the lack of suitable training data for robotics [3]. We argue instead that data relevant to training robotic manipulation models is copious but has important structure that most existing training methods ignore and fail to leverage. These insights lead us to propose a novel transformer-based architecture, called *PLEX*, that is capable of effective learning from realistically available robotic manipulation datasets.

We observe that robotics-relevant data falls into three major categories: **(1)** Video-only data, which contain high-quality and potentially description-annotated demonstrations for an immense variety of tasks but have no explicit action information for a robot to mimic; **(2)** Data containing matching sequences of percepts *and actions*, which are less plentiful than pure videos and don't necessarily correspond to meaningful tasks [4], but capture valuable correlations between a robot's actions and changes in the environment and are easy to collect on a given robot; **(3)** Small sets of high-quality sensorimotor demonstrations for a target task in a target environment. Thus, a scalable model architecture for robotic manipulation must be able to learn primarily from videos, while being extra data-efficient on sensorimotor training sequences and the small amount target demonstrations.

PLEX, the **PL**anning-**EX**ecution architecture we propose, is designed to take advantage of data sources of these types. A PLEX model has two major transformer-based components: **(I)** a task-conditioned observational *planner* that, given a task specification and an estimate of the current

---

[*]Stanford University, gwthomas@stanford.edu. Work done partly while at Microsoft Research.
[†]Microsoft Research, {chinganc,riloynd,fevieira,vivineet,akolobov}@microsoft.com
[‡]dexman.ai, mihai@dexman.ai. Work done partly while at Microsoft Research.

world state, determines the next state to which the robot should attempt to transition, and **(II)** an *executor* that, having received the desired next state from the planner, produces an action that should lead there from the current state. The executor is trained by optimizing an inverse dynamics loss over exploratory sensorimotor data of the aforementioned category **(2)**, while the planner is trained by minimizing a loss of its autoregressive predictions computed with respect to video-only trajectories of category **(1)**. The target-task data of category **(3)** can be optionally used to efficiently finetune the planner, the executor, or both.

We make three design choices that greatly help the data efficiency of PLEX's training:

- *Learning to plan in the observation embedding space.* Rather than generating videos of proposed task execution using, e.g., stable diffusion as in Du et al. [5], PLEX learns to plan and execute in the low-dimensional space of observation embeddings.

- *Asymmetric learning of the embedding space.* The observation embedding space in which the executor and the planner operate is induced by training the observation encoder using the executor's loss *only* (or even by employing a frozen feature-rich encoder such as R3M [6]). The planner's gradients don't affect the encoder, which reduces the cost of PLEX training.

- *Relative positional encodings.* We adopt the relative positional encodings [7] in PLEX. We empirically show that in robotic manipulation the relative positional encodings significantly improve training efficiency from human-collected data compared with the *absolute* positional encodings [1] commonly used in the literature on transformers.

Most approaches that use video-only demonstrations for pretraining in robotic manipulation produce purely visual representations (see, e.g., [6, 8–10]). The majority of algorithms that produce sensorimotor models need most or all of the video demonstrations to be accompanied by action sequences that generated the videos, a requirement that holds only for a small fraction available manipulation data [11–17]. Few approaches have a dedicated trainable planning component; e.g. [16, 18–21] plan in a skill space, which PLEX can be modified to do as well. Conceptually, PLEX falls under the paradigm of learning from observations (LfO), but existing LfO approaches don't have multitask zero-shot planning capability [22–25] or demostrate it only in low-dimensional environments across similar tasks [26]. Of the works that have used transformers for robotic manipulation [14, 17, 21, 27, 28], only Brohan et al. [17] have analyzed their data efficiency, and none have looked at positional embeddings as a way to improve it. Overall, the closest approach to PLEX is the concurrently proposed UniPi [5]. It also has counterparts of PLEX's planner and executor, but its planner operates using diffusion *in the image space* [29], which is expensive both datawise and computationally, and may fail to model manipulation-relevant 3D object structure consistently [29]. A more extensive discussion of prior work is provided in Appendix A.

We experimentally show that PLEX's planner-executor design can effectively exploit the structure of realistically available robotic manipulation data to achieve efficient learning. On the multi-task Meta-World [30] benchmark, despite pretraining mostly on video data, PLEX exhibits strong zero-shot performance on unseen tasks and can be further improved by finetuning on a small amount of video-only demonstrations. We empirically show on the challenging Robosuite/Robomimic [31, 32] benchmark that, contrary to conclusions from NLP [7], the use of relative positional encodings significantly improves the data efficiency of PLEX learning from human-collected demonstrations.

## 2 Problem statement and relevant concepts

### 2.1 Problem statement

We consider the problem of learning a generalist task-conditioned policy for goal-directed object manipulation. Namely, we seek a policy that can control a robotic manipulator to successfully accomplish tasks that the robot may not have encountered during the policy training process; such a policy formally can be viewed a solution to a task-conditioned partially observable Markov decision process (POMDP) described in Appendix B. In practice, learning a generalist policy that performs well on a broad distribution of tasks zero-shot is very challenging, as the coverage and amount

of publicly available training data are limited. Therefore, in this work we consider a two-phased learning process: (1) pretraining, during which a generalist policy is trained, and (2) finetuning, during which this policy is adapted to a target task.

## 2.2 Data for training robotic manipulation models

We consider three broad groups of datasets relevant to training robotic manipulation systems:[4]

**Multi-task video demonstrations ($\mathcal{D}_{\textbf{mtvd}}$).** Being the most abundant category, it comprises data collections ranging from general YouTube videos to curated benchmarks such as Ego4D [33], Epic Kitchens [34, 35], and YouTube-8M [36] showing *an* agent – either a robot or a person – performing a meaningful object manipulation task with an end-effector. This data contains demonstration-quality sequences of video observations and descriptions of tasks they accomplish, but not the action sequences whose execution generated these videos.

**Visuomotor trajectories ($\mathcal{D}_{\textbf{vmt}}$).** These trajectories consist of paired sequences of observations and robots' actions. Although some of them may be high-quality demonstrations of specific tasks, e.g., as in the Bridge Dataset [15], many of these trajectories are generated by activities that most people will not find meaningful, e.g., grabbing random objects in a tray, as in the RoboNet [4]. Since no strong quality, quantity, or task association requirements are imposed on $\mathcal{D}_{\text{vmt}}$ data, it is relatively easy to collect for any target embodiment and environment.

**Target-task demonstrations ($\mathcal{D}_{\textbf{ttd}}$).** This is the most scarce but also most desirable data category, since it encompasses high-quality trajectories for a specific task in question, ideally collected on the target embodiment (robot). Note, however that we don't require that these demonstrations be visuo-motor. In fact, our experiments show that PLEX needs only video demonstrations for finetuning to learn a high-quality policy for a target task.

**A key data assumption** we make in this work is that $|\mathcal{D}_{\text{ttd}}| \ll |\mathcal{D}_{\text{vmt}}| \ll |\mathcal{D}_{\text{mtvd}}|$.

## 2.3 Transformers and positional encodings

A transformer-based architecture consists of several specially structured *self-attention layers* and, in general, maps an input *set* (often called a *context*) of $K$ elements (called *tokens*) to an output of the same size $K$ [1]. In most applications, such as language translation, transformers need to map *ordered* sets (i.e. sequences) to other ordered sets, and therefore add special vectors called *positional encodings* to each input element to identify its position in a sequence. These encodings can be learned as part of transformer's training or be hand-crafted.

The most common scheme is the *absolute positional encoding*, where each position in the transformer's $K$-sized context gets a positional vector [1]. Some transformers, e.g., Chen et al. [37], use what we call a *global positional encoding*. It is similar to the absolute one, but assigns a separate vector to each position *in the entire input sequence* rather than just the $K$-sized context, up to some maximum length $T \gg K$. Finally, models based on Transformer-XL [7, 14, 17], instead condition the attention computation on the *relative* positions between different pairs of input tokens within a context. In this work, we argue that on robotic manipulation finetuning datasets that consist of small numbers of human-gathered demonstrations, relative positional encoding is significantly more data-efficient than absolute or global one.

## 3 PLEX architecture and training

### 3.1 Intuition

PLEX (shown in Figure 1) separates the model into two transformer-based submodules: *1)* a *planner* that plans in the observation embedding space based on a task specification, and *2)* an *executor* that takes the embeddings of the historical and the planned future observations and outputs an action to control the robot.

---

[4]Static image datasets, e.g., ImageNet, aren't treated by PLEX in a special way and we don't discuss it here, but can be used to pretrain PLEX's image encoders.

This design is motivated by the structure of $\mathcal{D}_{\text{mtvd}}$, $\mathcal{D}_{\text{vmt}}$, and $\mathcal{D}_{\text{ttd}}$ dataset categories, which as we explain below make them suitable for three complementary learning objectives.

1. **Learning to execute state transitions.** The visuomotor trajectories from $\mathcal{D}_{\text{vmt}}$, collected on the target robotic manipulator or a similar one, show the robot how to execute a wide variety of state transitions. By sampling an observation-action tuple $\langle o_{t-H}, \ldots, o_t, a_t, o_{t+L} \rangle$, the agent can learn to infer $a_t$ from $o_{t-H}, \ldots, o_t$, and $o_{t+L}$ using *inverse dynamics*, where $t$ is the current time step, $H$ is an observation history length, and $L$ is a lookahead parameter.

2. **Learning to plan for tasks.** In order to recommend a meaningful action at each step, inverse dynamics inference needs the (embedding of) the desired future observation. Determining the desired future observation *given a task description* is something that can be learned from multi-task video-only data $\mathcal{D}_{\text{mtvd}}$, since this data shows what progress towards a successful completion of a specified task should look like.

3. **Improving target-task performance.** While learning to plan and execute on diverse $\mathcal{D}_{\text{mtvd}}$ and $\mathcal{D}_{\text{vmt}}$ data can result in a robotic manipulation foundation model [3] with strong zero-shot performance (see Section 4.2), on many tasks it may be far from perfect. Small datasets $\mathcal{D}_{\text{ttd}}$ of high-quality target-task demonstrations (e.g., through teleoperation) can provide additional grounding to the target domain to further improve a pretrained model.

### 3.2 Architecture

Following the above intuitions, we train PLEX's executor using data $\mathcal{D}_{\text{vmt}}$ and PLEX's

Figure 1: **PLEX architecture.** This diagram illustrates the information flow during PLEX training, described in Section 3.2. PLEX is optimized using the planner's loss $\mathcal{L}_{PL}$ (computation shown with black arrows ↑), and the executor's loss $\mathcal{L}_{EX}$ (computation shown with gray arrows ↑). The symbols '=' and '=' denote stopgrads, where backpropagation is halted. Each input modality $m$ is embedded using a modality-specific encoder $\phi_m$. Video demonstration embeddings $\tilde{g}, \tilde{I}_{1:T}$, and (optionally) $\tilde{R}_{1:T}$ are used to train the planner *over the embedding space* using the prediction loss $\mathcal{L}_{PL}$. Visuomotor trajectory embeddings $\tilde{I}_{1:T}, \tilde{p}_{1:T}, \tilde{a}_{1:T}$ are passed to the executor to compute the inverse dynamics loss $\mathcal{L}_{EX}$. Note that if the image encoder $\phi_I$ isn't frozen, $\mathcal{L}_{EX}$'s gradients will update $\phi_I$. In contrast, the planner's own loss $\mathcal{L}_{PL}$ never affects $\phi_I$ (see stopgrad symbol =).

planner using data $\mathcal{D}_{\text{mtvd}}$, in addition to a small dataset $\mathcal{D}_{\text{ttd}}$ of target-task trajectories (which, if available, can be used to train both the planner and executor). Specifically, let $\tau = g, R_1, I_1, p_1, a_1 \ldots, R_T, I_T, p_T, a_T = g, R_{1:T}, I_{1:T}, p_{1:T}, a_{1:T}$ denote a trajectory. Here, $g$ is a task specification, $I_t$ is a tuple of camera image observations, $p_t$ is a proprioceptive state, $a_t$ is an action, and $R_t$ is a return-to-go at time $t$, i.e. $R_t = \sum_{t'=t}^{T} r_{t'}$, where $r_{t'}$ is the instantaneous reward at time $t'$. The length $T$ can vary across trajectories. As Figure 1 shows, PLEX processes these input modalities using corresponding encoders $\phi_g, \phi_I, \phi_p, \phi_a$, and $\phi_R$ to obtain an embedded sequence $\tilde{g}, \tilde{R}_{1:T}, \tilde{I}_{1:T}, \tilde{p}_{1:T}, \tilde{a}_{1:T}$. When a modality is missing, it is replaced by trainable placeholder vectors during embedding. Missing modalities are common in robotic manipulation datasets; e.g., few datasets have rewards. Since PLEX's executor and planner are designed to be trainable on task-agnostic visuomotor $\mathcal{D}_{\text{vmt}}$ data and task-conditioned video-only demonstrations $\mathcal{D}_{\text{mtvd}}$, respectively, each of these components is specialized to operate only on the (embeddings of) modalities available in their prevalent training data. Per Figure 1, task description and return embeddings $\tilde{g}$ and

$\tilde{R}_{1:T}$ don't get routed to the executor, since they are missing from $\mathcal{D}_{\text{vmt}}$ data. Similarly, the planner only receives $\tilde{g}$, $\tilde{I}_{1:T}$ and, optionally, $\tilde{R}_{1:T}$ embeddings, since they are present in $\mathcal{D}_{\text{mtvd}}$ data. This separation holds also at the deployment time, when all modalities are available.

**Planner** The planner's sole purpose is to determine *where* the agent should go in the observation embedding space. As shown in Figure 1, given embeddings $\tilde{g}$, $\tilde{I}_{1:T}$ of a task-conditioned video-only training demonstration, the planner outputs a sequence $\hat{I}_{1+L:T+L}$ of embeddings corresponding to the observations the agent should ideally see $L$ steps in the future from its current time step; $L$ is a hyperparameter. The planner's training minimizes the prediction loss

$$\mathcal{L}_{PL}(\tilde{g}, \tilde{R}_{1:T}, \tilde{I}_{1:T}) = \sum_{t=1+L}^{T+L} \|\tilde{I}_t - \hat{I}_t\|_2^2. \tag{1}$$

where we set $\tilde{I}_t = \tilde{I}_T$ for $t = T+1, ..., T+L$. Crucially, $\mathcal{L}_{PL}$'s gradients *don't backpropagate* into the encoders $\phi_g$ and $\phi_I$. This is to prevent the collapse of the image embedding space (denoted as $\mathcal{E}_o$); note the stopgrad symbols on $\mathcal{L}_{PL}$'s computation paths in Figure 1. The embedding space $\mathcal{E}_o$ either comes from pretrained encoders or is learned with inverse dynamics during executor training.

**Executor** Like the planner, the executor has a specific role at the deployment time. Given the observation-action sequence $o_{1:t}, a_{1:t}$ so far and the target observation embedding $\hat{I}_{t+L}$ produced by the planner, the executor infers an action $\hat{a}_t$ for the current step. This inference step should be done in a task-agnostic way, as the task knowledge is already incorporated in the $\hat{I}_{t+L}$ prediction of the planner. For a trajectory from $\mathcal{D}_{\text{vmt}}$, we optimize the executor via the inverse dynamics loss

$$\mathcal{L}_{EX}(I_{1:T}, p_{1:T}, \hat{I}_{1+L:T+L}, a_{1:T},) = \sum_{t=1}^{T-1} \|a_t - \hat{a}_t\|_2^2 \tag{2}$$

A major difference between $\mathcal{L}_{EX}$ and $\mathcal{L}_{PL}$ optimization is that the former's gradients can backpropagate into the encoders $\phi_I$, $\phi_o$, $\phi_p$, and $\phi_a$: the computation path for $\mathcal{L}_{EX}$ through these encoders in Figure 1 doesn't have a stopgrad. This allows executor training to shape the embedding space $\mathcal{E}_o$.

**Relative positional encoding** Like the Decision Transformer (DT) [37], PLEX's planner and executor transformers are derived from GPT-2. However, DT's use of global positional encoding implicitly assumes that all training trajectories have the same length $T$. PLEX, in contrast, uses relative encoding from Dai et al. [7] as the default. As we show empirically, in robotic manipulation settings where tasks are usually goal-oriented and training demonstrations vary a lot in length, global positional embedding performs poorly and even the fixed absolute positional encoding common in NLP [1] performs much better. Especially, for human-collected demonstrations where variability is significant, our experimental results show that relative encoding [7] perform significantly better.

### 3.3 Training PLEX

Training PLEX generally involves both pretraining and finetuning, though the experiments in Section 4.2 show that pretraining alone already gives PLEX solid zero-shot performance.

**Pretraining** PLEX consists of two sub-stages:

*1. Pretraining the executor* by optimizing the $\mathcal{L}_{EX}$ loss (Equation (2)) over a $\mathcal{D}_{\text{vmt}}$ dataset.

*2. Pretraining the planner* by optimizing the $\mathcal{L}_{PL}$ loss (Equation (1)) over a $\mathcal{D}_{\text{mtvd}}$ dataset.

If the observation encoders are expected to be trained or finetuned by the inverse dynamics loss $\mathcal{L}_{EX}$, rather than pretrained and frozen beforehand, it is critical for executor pretraining to be done before training the planner. Indeed, the planner is expected to make predictions in the observation encoders' embedding space, which will change if the inverse dynamics loss affects the encoders. If the encoders are frozen from the start, however, the pretraining stages can proceed asynchronously.

**Finetuning** involves adapting PLEX using a target-task demonstration dataset $\mathcal{D}_{\text{ttd}}$. As with any finetuning, this involves deciding which part of PLEX to adapt.

Since $\mathcal{D}_{\text{ttd}}$ can be viewed both as a small $\mathcal{D}_{\text{mtvd}}$ and a small $\mathcal{D}_{\text{vmt}}$ dataset, it can be used to train any component of PLEX—executor, planner, and observation encoders. As with pretraining, if $\mathcal{D}_{\text{ttd}}$ is used for finetuning the encoders, it is critical to complete their finetuning before finetuning the

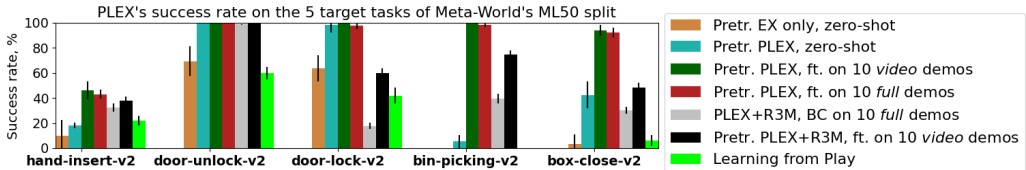

Figure 2: PLEX's generalization experiments. The confidence intervals are computed with 10 seeds.

planner. In Section 4.2, we show that finetuning just the last layer of the planner's transformer, which constitutes 5% of the parameters of the PLEX instance in the experiment, is sufficient for significantly boosting a pretrained PLEX's performance.

$\mathcal{D}_{\text{ttd}}$ can also be employed for optimizing a behavior cloning loss $\mathcal{L}_{BC}$. This amounts to training the planner, executor, and encoders *simultaneously* by having PLEX predict $\mathcal{D}_{\text{ttd}}$ trajectories's actions from the same trajectories' observations, and allowing the action prediction loss gradients to backpropagate through the entire PLEX model, to its the inputs. The experiments in Section 4.3 demonstrate the efficiency of BC-based finetuning thanks to the use of a relative position encoding.

## 4 Experiments

We conduct two sets of experiments to answer the following questions: *(i) Does PLEX pretrained on task-agnostic sensorimotor data and task-annotated video data generalize well to downstream tasks? (ii) How does the use of relative positional encodings affect PLEX's policy quality?* Appendix C provides the details about our PLEX implementation.[5]

### 4.1 Benchmarks and training data

**Meta-World:** Meta-World [30] is a collection of 50 tasks featuring a Sawyer arm. We use Meta-World-v2 with image observations (see details in Appendix D.1). We consider the ML45 split consisting of 45 training and 5 target tasks (***door-lock***, ***door-unlock***, ***hand-insert***, ***bin-picking***, and ***box-close***). We use these 5 target tasks for evaluation. Meta-World comes with high-quality scripted policies for all tasks. To get ***video demonstration data*** ($\mathcal{D}_{\text{mtvd}}$), we use these scripted policies to generate 100 successful video-only demonstrations for each of the 45 training tasks, i.e., $|\mathcal{D}_{\text{mtvd}}| = 4500$. To generate ***visuomotor trajectories*** ($\mathcal{D}_{\text{vmt}}$), for the 5 target tasks' environments, we add zero-mean Gaussian noise with standard deviation $0.5$ to the actions of the scripted policies and record the altered actions. We collect 50 trajectories per task, i.e., $|\mathcal{D}_{\text{vmt}}| = 250$. Finally, for ***target-task demonstrations*** ($\mathcal{D}_{\text{ttd}}$), we employ the original scripted policies to produce 75 demonstrations per target task and sample 10 of them in a finetuning experiment run, i.e., $|\mathcal{D}_{\text{ttd}}| = 10$.

**Robosuite:** Robosuite benchmark [31], compared Meta-World, has robotic manipulation tasks with a significantly more complicated dynamics and action space. We use 9 of its tasks involving a single robot arm (Panda) (***Lift***, ***Stack***, ***Door***, ***NutAssemblyRound***, ***NutAssemblySquare***, ***PickPlace-Bread PickPlaceCan***, ***PickPlaceMilk***, and ***PickPlaceCereal***). Robosuite's details are provided in Appendix D.1. Importantly, the training data for Robosuite was collected from human demonstrations, *not* generated by scripted policies as in Meta-World. See Appendix D.4 for details.

### 4.2 Generalization experiments

Here we focus on pretraining PLEX with multi-task Meta-World data. The results are shown in Figure 2. We train a 16,639,149-parameter PLEX instance (including the ResNet-18-based image encoder) from scratch with random initialization. We use the success rate on the 5 target tasks as the performance metric. For baselines, we experiment with PLEX with a frozen ResNet-50-based R3M [6], an observational representation pretrained on the large Ego4D dataset [33]. We denote it as *PLEX+R3M*; in Figure 2, *Pretr. PLEX+R3M* was first pretrained on multitask data and then finetuned on a target task, while *PLEX+R3M, BC* was trained only on a single target task's data from the start. In addition, we use an adapted *Learning from Play* (LfP) approach [11]. The

---

[5]We implement PLEX using the GPT-2 of the DT codebase [37] but without return conditioning.

hyperparameters and details can found Appendices C and D. In summary, the experimental results show that PLEX can perform well without seeing a single sensorimotor expert demonstration.

**PLEX demonstrates zero-shot generalization capabilities**    Figure 2 shows that PLEX pretrained on as few as 4500 video demonstrations ($\mathcal{D}_{\text{mtvd}}$) from the training environments and 250 dynamics trajectories ($\mathcal{D}_{\text{vmt}}$) from the target environments (denoted as *Pretr. PLEX, zero-shot* in Figure 2) exhibits good downstream performance *zero-shot*. To demonstrate that this performance is really due to planning learned from video-only data as opposed to the executor inadvertently exploiting biases in the data, we consider a PLEX variation (denoted as *Pretr. EX only, zero-shot*) where we only pretrain the *executor* (on $\mathcal{D}_{\text{vmt}}$), not the planner.[6] The results of *Pretr. EX only, zero-shot* reflect a level of performance one can get with knowledge contained in the dynamics data $\mathcal{D}_{\text{vmt}}$ alone. *Pretr. EX only, zero-shot* underperforms *Pretr. PLEX, zero-shot*, which shows the importance of learning from $\mathcal{D}_{\text{mtvd}}$ via PLEX's planner.

Our main baseline for zero-shot generalization is *Learning from Play* (LfP) [11], one of the few existing methods able to generalize zero-shot from data as low-quality as $\mathcal{D}_{\text{mtvd}}$. LfP has planning capability but doesn't have a way to use either the video-only data $\mathcal{D}_{\text{mtvd}}$ or the target-task demonstrations $\mathcal{D}_{\text{ttd}}$, and performs which gives PLEX a large advantage.

**PLEX can be finetuned effectively using only a few video-only demonstrations**    We further show that finetuning only 5% of PLEX's parameters (the last transformer layer of the planner) on just 10 *video-only* demonstrations for a given task significantly boosts PLEX's success rate there. For all 5 downstream tasks, this policy outperforms *Pretr. EX only, zero-shot* by $\geq 2\times$. The improvement is drastic especially in the case of *hand-insert-v2*, *bin-picking-v2*, and *box-close-v2*.

**Video-only demonstrations is all PLEX needs during finetuning**    Interestingly, we find that full demonstrations (with both video and action sequences) don't increase PLEX's performance beyond video-only ones. This can seen from the experimental results of *Pretr. PLEX, ft. on 10 full demos*, where we finetune PLEX (the action head and last transformer layer of PLEX's planner, executor; $\approx 11\%$ of PLEX) on 10 *full* (sensorimotor) demonstrations for each task. We think this is due to PLEX's image encoder being pretrained only on observations from $\mathcal{D}_{\text{vmt}}$ and frozen during finetuning. Because of this, finetuning couldn't help the encoder learn any extra features for modeling inverse dynamics *over the observation space region covered by $\mathcal{D}_{\text{ttd}}$*, even if such features would improve PLEX's performance.

The issue of impoverished observation coverage in $\mathcal{D}_{\text{vmt}}$ dataset can be addressed by using a frozen encoder pretrained on an independent large dataset, as the results of *PLEX+R3M, BC* and of *pretrained PLEX+R3M* in Figure 2 suggest. There, PLEX's R3M encoder was never trained on *any* Meta-World observations but enables PLEX to perform reasonably well.

The results of *Pretr. PLEX+R3M* and *PLEX+R3M, BC* in Figure 2 illuminate two other aspects of using observation-only representations like R3M: (1) The sensorimotor representation that PLEX learns *on top of* R3M clearly helps generalization – *pretrained PLEX+R3M* performs much better than *PLEX+R3M, BC*, which was trained only on a single task's data, despite *pretrained PLEX+R3M* seeing just video-only demonstrations at finetuning. (2) Fully frozen R3M somewhat limits PLEX's performance – PLEX variants that pretrained their own encoder outperform PLEX+R3M on 3 of 5 tasks.

### 4.3 Positional encoding experiments

In the Meta-World experiments, all training data was generated by scripted policies. In real settings, most such data is generated by people teleoperating robots or performing various tasks themselves. A hallmark of human-generated datasets compared to script-generated ones is the demonstration variability in the former: even trajectories for the same task originating in the same state tend to be different. In this section, we show that in low-data regimes typical of finetuning on human-generated

---

[6] At run time we feed the embedding of the task's *goal* image as the predictions that the *executor* conditions on (since no planner is trained).

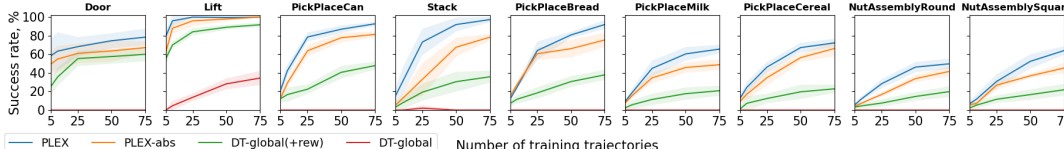

Figure 3: Data efficiency of PLEX's relative positional encoding in single-task mode on Robosuite's single-arm tasks with $|\mathcal{D}_{\text{ttd}}|$ varying from 5 to 75. **PLEX** (with relative encodings) in most cases significantly outperforms and at worst matches the performance of its version **PLEX-abs** with absolute positional encodings. Both versions significantly outperform DT.

demonstrations, PLEX with relative positional encoding yields superior policies for a given amount of training data than using absolute encoding. The results are in Figure 3.

**Baselines, training and evaluation protocol.** To analyze data efficiency and compare to prior results on Robosuite, we focus on an extreme variant of finetuning – training from scratch. For each of the 9 Robosuite tasks and each of the evaluated encodings, we train a separate 36,242,769-parameter PLEX instance using only that task's $\mathcal{D}_{\text{ttd}}$ dataset of full sensorimotor human-generated demonstrations. We compare PLEX with relative positional encoding to PLEX with absolute one and to two flavors of the Decision Transformer (DT) [37], which use global positional embedding. Appendix D.5 and Figure 3 provide more details about model training dataset collection, and the baselines. For each task/dataset size/approach, we train on 10 seeds.

**Results.** As Figure 3 shows, PLEX learns strong policies using at most 75 demonstrations, despite having to train a 36M-parameter model including randomly initialized vision models for tasks, most of which have complex dynamics and broad initial state distributions. Moreover, PLEX with relative positional encoding (denoted simply as *PLEX* in the legend) outperforms the alternatives by as much as 20 percentage points (pp) on Robosuite's human-generated demonstration data while never losing to them. In particular, *DT-global(+rew)* and, especially, *DT-global* perform far worse of both *PLEX* and *PLEX-abs*. Since all models share most of the implementation and are trained similarly when *PLEX* and *PLEX-abs* run in BC mode, we attribute PLEX's advantage only to the combined effect of using human-generated training data and positional encodings. We have also trained PLEX and PLEX-abs for Meta-World's 5 target tasks from the previous experiment for various amounts of the available – scripted – demonstrations for these tasks and noticed no significant performance difference between PLEX and PLEX-abs on any task. This provides additional evidence that the utility of relative positional enconding manifests itself specifically on human-generated demonstration data.

In fact, relying on relative positional encoding allows PLEX to achieve state-of-the art performance on all Robosuite tasks in this experiment, as we show and analyze empirically in Appendix D.4.

## 5 Conclusion and limitations

We have introduced PLEX, a transformer-based sensorimotor model architecture that can be pre-trained on robotic manipulation-relevant data realistically available in quantity. Our experimental results show that PLEX demonstrate strong zero-shot performance and can be effectively finetuned with demonstrations to further boost its performance. In particular, PLEX shows superior performance on human-collected demonstrations because of its usage of relative positional encoding.

**Limitations** We believe that PLEX has great potential as a model architecture for general robotic manipulation, but in most of our experiments so far, the training data came from the same robot on which the trained model was ultimately deployed. In reality, most available multi-task video demonstration data $\mathcal{D}_{\text{mtvd}}$ is generated by other robots or even people. This can cause a mismatch between the demonstrations and the target robot's capabilities and setups. Planning hierarchically first in the skill space as, e.g., in Lynch et al. [38], and then in the observation embedding space may address this issue. In addition, so far we have trained PLEX on simulated data. The eventual goal, and indeed a significant motivation for this work, would be to pretrain on internet-scale "in-the-wild" video datasets [29, 33, 36]. Also, with the rise of powerful LLMs such as Ouyang et al. [39], switching PLEX to language for task specification can facilitate generalization across tasks.

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

# Appendix

## A  Related work

Our work lies at the intersection of scalable multi-task representation learning for robotic manipulation, learning from observations, and decision-making using transformers.

**Representation learning for robotic manipulation.** Most approaches of this kind focus on pre-training purely *non-motor*, usually visual, representation models (see, e.g., [6, 8, 9, 40, 41], and references therein). These models don't output actions; they are meant to be foundations on top of which a policy network is to be learned. Thus, in contrast to PLEX, by themselves they can't enable zero-shot generalization to unseen tasks even in the limit of pretraining data coverage and amount. However, they are synergistic with PLEX: PLEX can use them as frozen observation encoders, as we show in Section 4.2 on the example of R3M [6].

Techniques that train sensorimotor models – i.e., full-fledged generalist policies, like PLEX – have also been rising in prominence. Some of them [42–44] are based on *meta learning* [45]. However, Mandi et al. [46] have shown multi-task pretraining followed by finetuning to be more effective when the task distribution is broad, and several approaches [11–17] follow this training paradigm as does PLEX. At the same time, most of them need pretraining data consisting of high-quality demonstrations in the form of matching videos *and* action sequences. While the quality requirement can be relaxed using offline RL, as, e.g., in Singh et al. [47], in order to enable generalization across broad task distributions these sensorimotor training demonstrations need correspondingly broad task coverage. This assumption is presently unrealistic and ignores the vast potential of the available video-only data — the weakness PLEX aims to address.

Among the sensorimotor representation learning methods that, like PLEX, try to learn from both video-only and sensorimotor data are Schmeckpeper et al. [48], Lynch and Sermanet [11], and Mees et al. [21]. Schmeckpeper et al. [48] consider single-task settings only and require the video-only and sensorimotor data to provide demonstrations for the same tasks. Lynch and Sermanet [11] and Mees et al. [21] allow the sensorimotor data to come from exploratory policies rather than task demonstrations but insist that this data must be generated from meaningful *skills*, a strong assumption that PLEX avoids.

Architecturally, most aforementioned approaches use monolithic models that don't have separate components for planning and execution like PLEX. Notable exceptions are methods that mine skills from pretraining data, embed them into a latent space, and use the latent skill space for accelerated policy learning of new tasks after pretraining [16, 18–21]. This is akin to planning in the skill space. PLEX can accommodate this approach hierarchically by having, e.g., a CVAE-based high-level planning model [38] produce a task-conditioned sequence of skill latents and feeding them into a skill-conditioned planning model that will plan in the observation embedding space. However, in this work's experiments, for simplicity PLEX plans in the observation embedding space directly.

**Learning and imitation from observations (I/LfO)** I/LfO has been used in robotic manipulation both for single-task tabula-rasa policy learning [22, 23] and pretraining [24]. Pathak et al. [24] is related to PLEX in spirit but lacks a counterpart of PLEX's planner. As a result, it can't complete an unseen task based on the task's goal description alone: it needs either a sequence of subgoal images starting at the robot's initial state or a sequence of landmarks common to all initial states of a given task. Beyond robotics, a type of LfO was also employed by Baker et al. [25] and Venuto et al. [49] to pretrain a large sensorimotor model for Minecraft and Atari, respectively. This model, like Pathak et al. [24]'s, doesn't have a task-conditioned planning capability and is meant to serve only as a finetunable behavioral prior. Xu et al. [26] investigate an LfO method akin to PLEX in low-dimensional environments, where it side-steps the question of choosing an appropriate representation for planning, the associated efficiency tradeoffs, and pretraining a generalizable planning policy.

Overall, the closest approach to PLEX is the concurrently proposed UniPi [5]. It also has a universal planner meant to be pretrained on a large collection of available videos, as well as an executor that captures inverse dynamics. However, UniPi ignores the issue of data efficiency and plans in the space of images (observations), using diffusion [29], rather than in the latent space of their embeddings. This is expensive to learn and potentially detrimental to plan quality. Latents even from statically pretrained image encoders are sufficient to capture object manipulation-relevant details from videoframes [8], whereas diffusion models can easily miss these details or model their 3D structure inconsistently [29]. Indeed, despite being conceptually capable of closed-loop control, for computational efficiency reasons UniPi generates open-loop plans, while PLEX interleaves planning and execution in a closed loop.

**Transformers for decision making and their data efficiency.** After emerging as the dominant paradigm in NLP [2] and CV [50], transformers have been recently applied to solving general long-horizon decision-making problems by imitation and reinforcement learning [37, 51–54], including multi-task settings [55] and robotic manipulation [14, 17, 21, 27, 28]. Mees et al. [21] provide evidence that in robotic manipulation transformers perform better than RNNs [11] while having many fewer parameters. Of all these works, only Reed et al. [14] uses relative positional encoding, and only by "inheriting" it with the overall Transformer-XL architecture [7], without motivating its effectiveness for decision-making.

**Task specification formats.** Task specification modality can significantly influence the generalization power of models pretrained on multi-task data. Common task conditioning choices are images of a task's goal [15], videos of a task demonstration by a person [12, 42] or by a robot [13, 45], and language descriptions [11, 12, 17, 21, 56]. PLEX is compatible with any of these formats; in the experiments, we use goal images.

# B    Problem formalization

Formally, the problem PLEX aims to solve can be described as a partially observable Markov decision process (POMDP) $\langle \mathcal{G}, \mathcal{S}, \mathcal{O}, z, \mathcal{A}, p, r \rangle$ with a special structure. Here, $\mathcal{G}$ is the space of possible manipulation tasks that we may want to carry out the tasks in $\mathcal{G}$. $\mathcal{S} = \mathcal{P} \times \mathcal{W}$ is a state space consisting of a space $\mathcal{P}$ of robots' proprioceptive states (e.g., poses, joint speeds, etc.) and a space $\mathcal{W}$ of world states. A state $s$'s proprioceptive part $p \in \mathcal{P}$ is known at execution time and in some of the training data, whereas the world state $w \in \mathcal{W}$ is never observable directly. A latent state $s$ can be probabilistically inferred from its observations $o \in \mathcal{O}$ and a state-conditioned distribution $z : \mathcal{S} \to \Delta(\mathcal{O})$ that describes how latent states in $\mathcal{S}$ manifest themselves through observations, where $\Delta$ denotes the space of distributions. For robotic manipulation, each observation can consist of several *modalities*: camera images (possibly from several cameras at each time step), depth maps, tactile sensor readings, etc. The distribution $z$ is unknown and needs to be learned. $\mathcal{A}$ is an action space, e.g., the space of all pose changes the robotic manipulator can achieve in 1 time step, and $p : \mathcal{S} \times \mathcal{A} \to \Delta(\mathcal{S})$ is a transition function describing how executing an action affects a current state, which potentially is stochastic. A reward function $r : \mathcal{G} \times \mathcal{S} \times \mathcal{A} \times \mathcal{S} \to \mathbb{R}$ can provide additional detail about task execution by assigning a numeric reward to each state transition, e.g., 0 for transitions to a task's goal state and -1 otherwise. Our objective is to learn a policy $\pi : \mathcal{G} \times \mathcal{O}_{|H} \to A$ that maps a history of observations $\mathcal{O}_{|H}$ over the previous $H$ steps to an action so as to lead the robot to accomplish a task $g \in \mathcal{G}$.

# C    PLEX implementation details

The transformers PLEX uses as its planner and executor are derived from the GPT-2-based version of the Decision Transformer (DT) [37]. Like in DT, we feed inputs into PLEX by embedding each modality instance (e.g., an image or an action) as a single unit. This is different to the way, e.g., Gato [14] and Trajectory Transformer [51] do it, by splitting each input into fragments such as image patches and embedding each fragment separately.

We condition PLEX's planner on embeddings of goal images. Low-dimensional inputs (actions and proprioceptive states) are mapped to $\mathbb{R}^h$, the transformer's $h$-dimensional input space, using a 1-layer linear neural network. High-dimensional inputs – videoframes from one or several cameras at each time step as well as goal images – are processed using a ResNet-18-based [57] encoder from Robomimic [32]. It applies a random crop augmentation to each camera's image, passes it through a separate ResNet18 instance associated with that camera, then passes the result through a spatial softmax layer [58], and finally through a small MLP. The resulting embedding is fed into PLEX's planner. If the robot has several cameras, the encoder has a separate ResNet instance for each. For each time step, PLEX's planner outputs an $h$-dimensional latent state representing the *predicted* embedding of PLEX's visual observations $k$ time steps into the future, where $k$ is a tunable parameter. These latents are then fed directly into the planner as predictions of future observation embeddings. The output latents from the planner transformer are fed through a $\tanh$ non-linearity, which outputs action vectors in the $[-1, 1]$ range. The hyperparameters can be bound in Tables 4 and 5.

Our PLEX implementation is available at https://microsoft.github.io/PLEX.

## D  Additional details about the experiments

### D.1  Meta-World and Robosuite details

**Meta-World.** In our Meta-World-v2 setup, at each time step the agent receives an $84 \times 84$ image from the environment's *corner* camera and the Sawyer arm's 18D proprioceptive state. The agent's actions have 4 dimensions, each scaled to the $[-1, 1]$ range. Although Meta-World also provides privileged information about the state of the environment, including the poses of all relevant objects, our PLEX agent doesn't access it.

**Robosuite.** The observation and action space in our experiments is exactly as in the best-performing high-dimensional setup from the Robomimic paper [32]. Namely, actions are 7-dimensional: 6 dimensions for the gripper's pose control (OSC_POSE) and 1 for opening/closing it. Visual observations are a pair of $84 \times 84$ images from *agentview* (frontal) and *eye-in-hand* (wrist) cameras at each step. Proprioceptive states consist of a 3D gripper position, a 4D quaternion for its orientation, and 2D gripper fingers' position.

### D.2  Details of the baselines from prior work

**PLEX +R3M [6].** We experiment with two combinations of PLEX with a frozen ResNet-50-based R3M [6], an observational representation pretrained on the large Ego4D dataset [33] In these experiments, R3M replaces Robomimic's ResNet-18, and we use versions of our Meta-World $\mathcal{D}_{\mathrm{vmt}}$, $\mathcal{D}_{\mathrm{mtvd}}$, and $\mathcal{D}_{\mathrm{ttd}}$ datasets with 224x224 image observations instead of the 84x84 ones.

One combination, *PLEX +R3M, BC* in Figure 2, learns a single-task policy on 10 full sensorimotor demonstrations for each Meta-World target task. It operates in behavior cloning (BC) mode, whereby PLEX is optimized solely w.r.t. its action predictions' MSE loss, whose gradients backpropagate though the whole network (except the frozen R3M). The other combination, *pretr. PLEX +R3M* in Figure 2, follows the same PLEX pretraining and finetuning process as described previously, except the R3M encoder stays frozen throughout.

**Learning from Play [11].** Our final baseline is an adapted *Learning from Play* (LfP) approach [11]. As in Lynch and Sermanet [11], LfP doesn't use video-only $\mathcal{D}_{\mathrm{mtvd}}$ data or target-task demonstrations $\mathcal{D}_{\mathrm{ttd}}$; it trains one model for all target tasks from the "play" dataset $\mathcal{D}_{\mathrm{vmt}}$ only. Instead of using language annotations to separate "meaningful" subsequences in $\mathcal{D}_{\mathrm{vmt}}$, we give LfP the ground-truth knowledge of where trajectories sampled from different tasks begin and end. Accordingly, we don't use language during training either. As n the case of PLEX, We train *Learning from Play* to plan conditioned only on goal images and present it with goal images from successful trajectories of the target tasks during evaluation.

### D.3 Success rate evaluation protocol

**In the generalization experiments on Meta-World**, all success rate evaluations are done on 50 500-step rollouts starting from initial states sampled from the *test* distributions of Meta-World's ML45 target tasks (*door-lock*, *door-unlock*, *hand-insert*, *bin-picking*, and *box-close*).

To evaluate the zero-shot success rate of the pretrained EX and PLEX models, we compute the average across 50 rollouts generated by these models on each of the 5 target tasks *at the end of pretraining*.

To evaluate the success rate of the finetuned models, we adopt the procedure from Mandlekar et al. [32]. The finetuning lasts for $N$ epochs (see Table 5). After each epoch, we measure the average success rate of the resulting model across 50 rollouts, and record the maximum average success rate across all finetuning epochs.

**In the positional encoding experiments on Robosuite**, the evaluation protocol is the same as in Meta-World finetuning and in Robomimic [32]: we train each model for $N$ epochs (see Table 5), after each epoch compute the success rate across 50 trajectories (with 700-step horizon), and record the best average success rate across all epochs.

### D.4 Robosuite datasets and model training

Training data for Robosuite was collected from human demonstrations, not generated by scripted policies. Robosuite provides a keyboard and SpaceMouse interfaces for controlling the Panda arm in its environments, and Robomimic supplies datasets of 200 expert ("professional-human") trajectories collected using the SpaceMouse interface for the *NutAssemblySquare*, *PickPlaceCan*, and *Lift* tasks. For each of the tasks without pre-collected Robomimic datasets, we gather 75 high-quality trajectories via Robosuite's keyboard interface ourselves. We employ Robosuite tasks only for experiments that involve training single-task policies from scratch, so all of these trajectories are used as ***target-task demonstration data ($\mathcal{D}_{ttd}$)***. Typical demonstration trajectory lengths vary between 50 and 300 time steps.

Accordingly, to show the difference between relative and absolute positional encodings' data efficiency, we train PLEX for $|\mathcal{D}_{ttd}| = 5, 10, 25, 50$, and 75, sampling $\mathcal{D}_{ttd}$'s from the set of 75 demonstrations without replacement. The results are presented in the main paper in Figure 3. For *Lift*, *PickPlaceCan*, and *NutAssemblySquare*, Robomimic [32] similarly provides 200 high-quality human-collected demonstrations each, as well as the results of BC-RNN on subsets of these datasets with $|\mathcal{D}_{ttd}| = 40, 100$, and 200. Therefore, for these problems we train PLEX for $|\mathcal{D}_{ttd}| = 5, 10, 25, 50, 75$, as well as $40, 100$, and 200. The results are shown in Table 3 and Table 1.

The only difference of PLEX model instances for Robosuite from those for Meta-World is the former having *two* ResNet-18s in the observation encoder, one for the eye-in-hand and one for the agentview camera. As for Meta-World, the encoder in the Robosuite is trained from scratch, in order to make our results comparable to Robomimic's [32], where models use an identical encoder and also train it tabula-rasa. In this experiment, we train PLEX in behavior cloning (BC) mode, like Meta-World's single-task PLEX +R3M, whereby PLEX is optimized solely w.r.t. its action predictions' MSE loss, whose gradients backpropagate though the whole network. All hyperparameters are in Table 5 in Appendix E.

We compare PLEX with relative positional encoding to PLEX with absolute one and to two flavors of the Decision Transformer (DT) [37], which use global positional embedding. One flavor (*DT-global* in Figure 3) is trained to condition only on task specification (i.e., goal images), like PLEX. We note, however, that Chen et al. [37] used rewards and returns when training and evaluating DT. Therefore, we also train a return-conditioned version of DT (*DT-global(+rew)* in Figure 3), with returns uniformly sampled from the range of returns in $\mathcal{D}_{ttd}$ during evaluation.

### D.5 Additional Robosuite results

**Comparison to BC-RNN.** Relying on relative positional encoding allows PLEX to achieve *state-of-the art* performance on all Robosuite tasks in our experiments. To establish this, in addition to the baselines in Figure 3, we compare to the results of a BC-RNN implementation from the work that introduced some of these Robosuite problems [32]. Interestingly, running BC-RNN on the tasks for which we have collected demonstrations ourselves resulted in 0 success rate (Table 2), while running it on tasks with Robomimic-supplied 200 trajectories (*Lift*, *PickPlaceCan*, and *NutAssemblySquare*) reproduced Mandlekar et al. [32]'s results. PLEX's comparison to BC-RNN's results on those problems are in Table 1 in Appendix D.4. PLEX and BC-RNN are at par on the easier problems but PLEX performs better on the harder *NutAssemblySquare*.

|  | *Lift* | | | *PickPlaceCan* | | | *NutAssemblySquare* | | |
|---|---|---|---|---|---|---|---|---|---|
| $|\mathcal{D}_{\text{ttd}}|$ | 40 | 100 | 200 | 40 | 100 | 200 | 40 | 100 | 200 |
| PLEX | $100 \pm 0$ | $100 \pm 0$ | $100 \pm 0$ | $82.8 \pm 8.9$ | $95.8 \pm 2.8$ | $96.6 \pm 4.1$ | $40.4 \pm 6.9$ | $69.6 \pm 4.1$ | $86.0 \pm 3.1$ |
| BC-RNN | $100 \pm 0$ | $100 \pm 0$ | $100 \pm 0$ | $83.3 \pm 1.9$ | $97.3 \pm 0.9$ | $98.0 \pm 0.9$ | $29.3 \pm 4.1$ | $64.7 \pm 4.1$ | $82.0 \pm 0.0$ |

Table 1: Performance of PLEX and BC-RNN on three Robosuite tasks from Mandlekar et al. [32] on $|\mathcal{D}_{\text{ttd}}| = 40, 100$, and 200 demonstrations. BC-RNN's results come from Figure 3b and Table 27 in Mandlekar et al. [32]). On the easier *Lift* and *PickPlaceCan*, PLEX and BC-RNN are at par, but on the harder *NutAssemblySquare* PLEX performs better. On the remaining 6 problems for which we have gathered the demonstration data, BC-RNN's success rate is 0 — see Table 2.

|  | *Door* | *Stack* | *PickPlaceBread* | *PickPlaceMilk* | *PickPlaceCereal* | *NutAssemblyRound* |
|---|---|---|---|---|---|---|
| $|\mathcal{D}_{\text{ttd}}|$ | 75 | 75 | 75 | 75 | 75 | 75 |
| PLEX | $78.4 \pm 9.2$ | $97.3 \pm 2.9$ | $92.0 \pm 4.65$ | $65.6 \pm 4.6$ | $72.2 \pm 4.4$ | $49.8 \pm 5.5$ |
| BC-RNN | $0 \pm 0$ | $0 \pm 0$ | $0 \pm 0$ | $0 \pm 0$ | $0 \pm 0$ | $0 \pm 0$ |

Table 2: Performance of PLEX and BC-RNN on the remaining 6 Robotsuite/Robomimic tasks from Figure 3. PLEX's numbers are copied from that Figure.

**Better data efficiency or higher performance?** Given Figure 3, one may wonder: does PLEX-abs's performance plateau at a lower level than PLEX's with relative positional encoding, or does PLEX-abs catch up on datasets with $|\mathcal{D}_{\text{ttd}}| > 75$? For most tasks we don't have enough training data to determine this, but Table 3 in Appendix D.4 provides an insight for the tasks with Robomimic-supplied 200 training demonstrations. Comparing the performance gaps between PLEX and PLEX-abs on 75-trajectory and 200-trajectory datasets reveals that the gap tends to become smaller. The same can be seen for *Stack*, *PickPlaceCereal*, *NutAssemblyRound* already at $|\mathcal{D}_{\text{ttd}}| = 75$ in Figure 3, suggesting that with sufficient data PLEX-abs may perform as well as PLEX. However, the amount of data for which this happens may not be feasible to collect in practice.

|  | *Lift* | | *PickPlaceCan* | | *NutAssemblySquare* | |
|---|---|---|---|---|---|---|
| $|\mathcal{D}_{\text{ttd}}|$ | 75 | 200 | 75 | 200 | 75 | 200 |
| PLEX | $100 \pm 0$ | $100 \pm 0$ | $80.4 \pm 5.7$ | $96.6 \pm 4.1$ | $64.0 \pm 4.6$ | $86.0 \pm 6.1$ |
| PLEX-abs | $100 \pm 0$ | $100 \pm 0$ | $72.8 \pm 8.0$ | $93.0 \pm 4.7$ | $45.2 \pm 5.7$ | $76.8 \pm 4.9$ |

Table 3: Performance of PLEX and PLEX-abs as the amount of training data $|\mathcal{D}_{\text{ttd}}|$ increases from 75 to 200 trajectories. The performance gap between the two is narrower on the larger dataset. For *Lift* and several other Robosuite tasks, this trend becomes visible for datasets smaller than 200 (see Figure 3.

## E  Hyperparameters

| Parameter name | Meta-World (*PLanner/EXecutor*) | Robosuite (*PLanner/EXecutor*) |
|---|---|---|
| # layers | 3/3 | 3/3 |
| context size $K$ | 30/30 time steps | 30/30 time steps |
| hidden dimension | 256/256 | 256/256 |
| # transformer heads | 4/4 | 4/4 |
| # evaluation episodes | 50 | 50 |
| # max. evaluation episode length | 500 | 700 |

Table 4: Hyperparameters of PLEX's transformer-based planner and executor components for the Meta-World and Robosuite benchmarks. In each case, the planner and executor use the same parameters, but for most problems the executor's context length $K$ can be much smaller than the planner's without loss of performance, e.g., $K_{EX} = 10$. For the Decision Transformer on Robosuite, we use 4 transformer layers and otherwise the same hyperparameters as for PLEX.

| Parameter name | Meta-World | | Robosuite |
|---|---|---|---|
| | pretraining (*PLanner/EXecutor*) | last-layer finetuning (*PLanner/EXecutor*) | behavior cloning (*PLanner/EXecutor*) |
| lookahead steps | $1/-$ | $1/-$ | $1/-$ |
| learning rate | $5 \cdot 10^{-4}$ | $5 \cdot 10^{-4}$ | $5 \cdot 10^{-4}$ |
| batch size | 256 | 256 | 256 |
| weight decay | $10^{-5}$ | $10^{-5}$ | $10^{-5}$ |
| # training epochs | 10/10 | 10/10(?) | 10 |
| # training steps per epoch | 250/250 | 250/250(?) | 500 |

Table 5: Hyperparameters of PLEX training for the generalization experiments on Meta-World and positional encoding experiments on Robosuite. The former use PLEX in pretraining and finetuning modes; the latter only in behavior cloning mode (training the entire model from scratch for a single target task). In finetuning mode, we adapt only the last transformer layer of the planner and, in one experiment, of the executor as well. The (?) next to the executor's hyperparameters indicate that they were used only in the experiment where the executor was actually finetuned. For the Decision Transformer on Robosuite we use the same hyperparameters as for PLEX.

