# OpenReview forum: "PLEX: Making the Most of the Available Data for Robotic Manipulation Pretraining"
_robot-learning.org/CoRL/2023/Conference — CoRL 2023 Poster_

### Official Review · Reviewer_aYEz · 2023-07-12

**Confidence:** 5
**Originality:** Good
**Technical Quality:** Good
**Clarity Of Presentation:** Excellent
**Impact:** 3

**Recommendation:**

Weak Reject: I recommend rejecting the paper, but will not argue for my recommendation if the majority of other reviewers have a different opinion.

**Review:**

Strengths
1. Very well written paper with regards to clarity and organization, easy to follow and understand.
2. Interesting idea that positively adds to the community's aspiration of building a foundation model for robotics.
3. Limitations are clearly stated.
4. Positional encoding ablation was an interesting finding.

Weaknesses
1. Baselines lacking: the method is not really compared to any baselines that also claim zero-shot generalization.
2. State-of-the-art claim not fully standing.
3. Mismatch of expectations: Zero-shot generalization can be of variant degrees of difficulty. The paper initially prepares the reader for learning from videos in-the-wild which didn't end up happening. In fact, the experiments that relied on encoders pretrained on human videos demonstrated worse performance which implies that the domain gap was left unaddressed. Ultimately, the method shows good results when video demonstrations and target tasks are in the same environment with the same embodiment which is a strictly easier type of zero-shot generalization to tackle.

**Quality Of The Limitations Section:**

Limitations are addressed clearly

**Questions For Rebuttal:**

1. Ln. 31: "robotic" --> "robot"
2. Ln. 32: "meaningful task" --> "meaningful tasks"
3. Ln 53-56: Asymmetric learning of the embedding space, re: I don't quite understand this. The space that the executor and planner operate in can be induced by a pretrained frozen encoder, yet you are doing learning. You mean this is just the backbone and gets fine-tuned? This space can alternatively be induced by the executor's training loss only. Why would the planner gradients increase the training cost so much and/or cause latent space collapse? Not obvious. It's possible that these may become clear later but at this point in the paper you have not yet described your training setup at all. Consider moving the design choices after you've presented your training setup.
4. Ln. 92-92: learning process is two-fold with second stage being finetuning, re: I was surprised to read this given that thus far, I was under the impression authors claim zero-shot generalization?
5. Ln. 106: "Since no strong quality, quantity, or task association requirements are imposed...", re: Why not? I would argue that if one is doing LfO, quality and quantity at least do matter very much.
6. Ln. 166-167: "While learning to plan and execute ... can result in a robotic manipulation foundation model with strong zero-shot performance", re: can it? Is this something that is shown later on in results? If yes, please add a note here to refer the reader to the results section. If not, please reduce the claim.
7. Ln. 178: "Rt is the reward-to-go at time t", re: since this is a term mostly appearing in RL and even there it's not seen as commonly, considering defining this with the formula to help readers.
8. The planner's gradients not flowing into encoders of g and I to prevent representation collapse is still not obvious. Could you please explain why this would necessarily cause a collapse?
9. Fig. 2: what does PLEX zero-shot mean for door-unlock when door-unlock in the ML50 split seems to be included in both training and testing sets according to: https://meta-world.github.io/ ?
10. Ln. 252-253: adding noise to the actions, re: this was not mentioned earlier. Is that just meant to be standard augmentation for robustness? Or serves some other purpose?
11. Ln. 286: "samll" --> "small"
12. finetuning only the last layer of the planning transformer being enough, re: may this suggest you are using a larger transformer than you need? Would like to see some experiments that investigate the impact of the planning transformer size as well as how does small transformer + finetuning the full model vs a large tranformer + finetuning only the last layer compare.
13. Ln. 306: pretrained PLEX + R3M performs much better than the single-task one, re: what does "single-task one" refer to here? I don't see anything single task in Fig. 2.
14. General comment: Looking at results, it looks like (and authors do mention this as well) PLEX + R3M where R3M is pretrained on Ego4D and frozen hurts generalization as opposed to PLEX still training all its components on the same domain (e.g. Meta World robot arm doing various tasks). This suggests that still the domain gap of learning from videos in-the-wild is still not really bridged and the method does not actually learn from videos in-the-wild but rather relies on pretrained encoders on such data as backbones to finetune. The introduction leaves the reader with the expectation that the method learns based on videos in-the-wild which is not really what's happening. Suggest reworking the message early on in the paper to avoid mismatch of expectations.
15. Appendix D5 - Comparison with BC-RNN: "On the remaining 6 problems for which
we have gathered the demonstration data, BC-RNN’s success rate is 0": Robomimic did not provide results on the other 6 robosuite tasks. Did you train Robomimic's BC-RNN as well as PLEX with the data from the remaining 6 tasks you collected? If yes, please add those as well so that we can see PLEX's performance compared to BC-RNN even if BC-RNN is all 0.
16. State-of-the-art is referenced multiple times in the paper. In Appendix D5, quoting "Relying on relative positional encoding allows PLEX to achieve state716 of-the art performance on all Robosuite tasks in our experiments.". Technically speaking, it does not perform better in all tasks. BC-RNN is slightly better on PickPlaceCan.
17. Baselines in main paper were unsatisfying. Only Decision Transformer was included which I wouldn't have considered the most suitable baseline for a one-to-one comparison. Please include comparisons with other methods that explicitly claim zero-shot generalization (e.g. https://arxiv.org/pdf/2011.05970.pdf).

**Robotics Focus:**

Highly relevant to robotics but no hardware experiments

**Summary Of Paper:**

This work proposes a modularized method that attempts to learn: a) visuomotor actions through task-agnostic videos of a robot (e.g. play data) and b) planning via predicting embeddings of future states in image space through task-conditioned object manipulation videos of tasks other than those used for evaluation. The two parts are trained on disjoint data. The goal is to move a step forward towards building a foundation model for robotic manipulation that can achieve zero-shot generalization at test time and can learn from arbitrary video demonstrations including human.

**Summary Of Recommendation:**

The method proposed was genuinely interesting, thank you to the authors for presenting. A direction like this could be promising towards eventually building a foundation model for robotics manipulation. My recommendation is weak reject at the moment primarily due to:

a) the paper is framing the work more about learning from videos in-the-wild where experiments stay in the same domain for both executor and planner models. Please rework the introduction to set expectations for the body of the paper accordingly.

b) the method is not being compared with other baselines that actually claim zero-shot generalization. Please add at least one baseline of a method that claims to be doing zero-shot generalization.

c) SOTA seems like too strong of a claim. Please do not claim SOTA since technically on one task, the method is not doing better and/or add numbers for the remaining 6 tasks.

I'm willing to increase my score if the suggested resolutions are addressed.

---

> ### Author Response · Authors · 2023-08-11
> **Rebuttal cont'd**
>
>
> 10. This is meant to simulate the situation where the visuomotor data $\mathcal{D}_{vmt}$ is of low quality and doesn’t come from the same trajectories as video-only data. Our intent was to demonstrate that PLEX does well under these conditions.
> This goes back to your question #5 – we believe that PLEX’s practical strength is exactly this ability to learn from low-quality visuomotor data as long as PLEX also has access to high-quality *video-only* demonstrations.
>
> 11. Fixed.
>
> 12. We tend to think that this means we are using a *sufficiently large* transformer. *Ideally* we would like our transformer to be large enough to avoid finetuning altogether, i.e., have zero-shot transfer to target tasks, but during our experiments we didn’t have a sufficient amount of data to train such a large model. Short of that, we would like our model to be sufficiently large/expressive to achieve strong target-task performance by finetuning as few of the model’s “logical units” as possible. In PLEX’s case, a “logical unit” is a transformer layer, so the model size in our experiments achieves this “sufficiency”.
>
> At some point we ran experiments of the type you mentioned, but we didn’t include their results in the paper and discarded them. We are rerunning them now, and will hopefully have their results available for you by August 15.
>
> 13. “Single-task” refers to “PLEX+R3M, BC” (see Figure 2 and l. 304 in the *original* submission), a baseline where the same PLEX architecture was trained using the BC loss *only* on demonstrations from the target task, without any pretraining. Sorry for the confusion, we have reworded this part and added clarifications at the start of Section 4.2 in the revised submission.
>
> 14. This is a fair observation – we have changed the introduction in the revised submission to avoid creating the impression that our experiments are using in-the-wild videos in this paper.
>
> 15. Yes, we trained Robomimic’s BC-RNN on the 6 Robosuite tasks ourselves. Moreover, as a sanity check, we also trained the BC-RNN on the Robomimic tasks, for which the results were provided by the Robomimic paper, and we were able to reproduce those results.
> We’ve added Table 2 to Appendix D.5 in the revised version with a comparison on the remaining tasks.
>
> 16. It is true that BC-RNN does slightly better on PickPlaceCan, but, as is common in the literature, by the SOTA claim we simply meant that PLEX outperforms or matches BC-RNN on the vast majority of Robosuite/Robomimic tasks we evaluated on, which is the case. As mentioned in #15, we have added full comparison results against BC-RNN to the Appendix.
>
> 17. While, admittedly, this was non-obvious from the experiments’ description, our main baseline for zero-shot evaluation was Learning from Play [17]. For zero-shot, it is the most directly comparable method to PLEX we are aware of, as it is one of the few existing methods capable of generalizing zero-shot from data as low-quality as $\mathcal{D}_{vmt}$. In the revised version, we have made this much more explicit.
>
> Please note also that the method you cited, https://arxiv.org/pdf/2011.05970.pdf, isn’t compararble to PLEX, as it claims a very specific type of ZS generalization, requires a full video demo to condition on at test time, and, most importantly, requires *expert sensorimotor* demonstrations for *pretraining*. Especially the latter is an example of unrealistic data requirement that PLEX is designed to avoid – indeed, in the experiments we don’t have such data for pretraining.

---

> ### Author Response · Authors · 2023-08-16
> **Rebuttal cont'd -- Experiments with PLEX size hyperparameters**
>
> This post is in response to question #12 in your review, asking for more experiments on PLEX's size and finetuning.
>
> **First, we show that we don't need a larger PLEX architecture instance to achieve strong post-finetuning performance on Meta-World (MW).**
>
> Let's call a PLEX architecture instance with $e$ executor transformer layers and $p$ planner transformer layers as a (e,p)-PLEX. Recall that the paper presents MW results on a (2,2)-PLEX. To demonstrate that (2,2)-PLEX is sufficient we train a (3,3)-PLEX and a (4,4)-PLEX using the same pretraining/finetuning protocol and on the same data as in the paper, and compare their success rates on the 5 target tasks of ML45 to those achieved by (2,2)-PLEX.
>
> **Please refer to the "PLEX hyperparameter tuning experiments" figure in https://anonymous68546.github.io/plex/**. The "(2,2)-PLEX, last-layer ft. on 10 $video$ demos" in this figure corresponds to "Pretr. PLEX ft. on 10 video demos" in Figure 2 in the paper. Compare its performance to the larger (3,3)-PLEX and (4,4)-PLEX. We see that (3,3)-PLEX's performance after last-planner-layer finetuning is the same as (2,2)-PLEX's, and the even larger (4,4)-PLEX's performance may even be slightly worse than (2,2)-PLEX's (possibly because (4,4)-PLEX is too large for the amount of data we used and for Meta-World's fairly low complexity, causing (4,4)-PLEX to overfit. Thus, we don't need to go larger than (2,2)-PLEX on Meta-World.
>
>
> **Second, we show that full finetuning of a smaller PLEX instance can work, but needs more data than finetuning just the last planner layer of a the (2,2)-PLEX instance used in the paper.**
>
> NOTE: The paper showed in Fig. 2 that for a sufficiently large PLEX architecture, finetuning just the last layer of the (video) planner is enough, so we can use *video-only* target-task demonstrations. In contrast, for full finetuning as we are doing in this small experiment, we necessarily have to use *fully visuomotor* target-task demonstrations, because finetuning the executor layers requires visuomotor data.
>
> Please refer again to the "PLEX hyperparameter tuning experiments" figure in https://anonymous68546.github.io/plex/. As smaller instances of PLEX, we experiment with (2,1)-PLEX and (1,2)-PLEX. We pretrain each of these instances following the paper's protocol on Meta-World, and then finetune the entire model using 10 and 25 demostrations from each target task.
>
> The results immediately show two things:
>
> 1. (2,1)-PLEX and (1,2)-PLEX finetuned *fully* using the same number of trajectories (10) that we used for finetuning just the planner layer of (2,2)-PLEX perform much worse than finetuned (2,2)-PLEX. This is not surprising: although (2,1)-PLEX and (1,2)-PLEX are smaller than (2,2)-PLEX, finetuning the small models in their entirety involves adapting many more parameters, and 10 trajectories is too little for that.
>
> 2. Finetuning smaller PLEX instances ((1,2)-PLEX, in particular) can in principle match finetuned (2,2)-PLEX's performance, but requires much more data (25 demos vs. 10).
>
> Thus, we conclude that finetuning just 1 layer of a larger PLEX instance is more data-efficient that fully finetuning smaller PLEX instances, and this is a major advantage of the former.

---

### Official Review · Reviewer_8ioC · 2023-07-15

**Confidence:** 4
**Originality:** Good
**Technical Quality:** Very Good
**Clarity Of Presentation:** Excellent
**Impact:** 4

**Recommendation:**

Strong Accept: I recommend accepting the paper and will argue for my recommendation even if other reviewers hold a different opinion.

**Review:**

Strengths

1. The paper provides an insightful analysis of groups of datasets relevant to robot learning based on quality, modality and volume. Thereafter the paper proposes an architecture and training methodology motivated by the above analysis.

2. Empirical results demonstrate that the proposed architecture is effective and data-efficient.

3. The approach is fairly novel and the writing is clear.

Weaknesses

1. I encourage the authors to add a baseline similar to 'Learning from play' but trained on sufficiently many sensorimotor demonstrations.
     1.1 In my understanding, the reason the 'Learning from play' baseline performs poorly is because of fewer sensorimotor demonstrations     available. While this setting is valid given the hypothesis presented in the dataset analysis, the results of the proposed baseline will further establish the data-efficiency of the approach.

2. I encourage the authors to perform experiments on demonstrations of long-horizon robot manipulation tasks. Will just fine-tuning the planner on long-horizon video demonstrations lead to good performance on such tasks? The authors should investigate this.

Minor revisions

1. Correct 'samll' to 'small' in line 286.


**Quality Of The Limitations Section:**

Limitations are addressed clearly

**Questions For Rebuttal:**

1. I encourage the authors to add a baseline similar to 'Learning from play' but trained on sufficiently many sensorimotor demonstrations. Refer to my comment in the Weakness list, point 1.

2. Please perform experiments on demonstrations of long-horizon robot manipulation benchmarks. Will just fine-tuning the planner on long-horizon video demonstrations lead to good performance on such tasks? I encourage the authors to investigate this.

**Robotics Focus:**

Highly relevant to robotics but no hardware experiments

**Summary Of Paper:**

The paper proposes an architecture for robot-learning for manipulation from readily available data. The paper conducts an analysis of the variety of data available and the volume of each type, identifying that unsupervised video demonstrations are easier to collect than sensorimotor data followed by task-specific data. Motivated by this, the paper presents a novel modular architecture and training paradigm.

The architecture is composed of a planner and an executor. First, the planner is trained using unsupervised video-demonstrations and then the executor is trained using sensorimotor data. Since observation (image) encoders are shared by the planner and the executor, the executor can be trained with fewer demonstrations (on top of frozen image encoders) after the planner is trained.  The architecture can also be fine-tuned using task-specific demonstrations.

Experiments provide good empirical demonstration of the efficacy and data-efficiency of the approach.

**Summary Of Recommendation:**

I did not give a lower rating becasue:

1. The dataset analysis presented by the authors is insightful. The architecture is devised keeping in mind the nature of available datasets.

I did not give a higher rating because:

1. Experimental results can be richer, solving more complex long-horizon tasks.

2. Hardware experiments are unavailable.

---

> ### Author Response · Authors · 2023-08-16
> **Rebuttal cont'd -- extra Learning from Play experiments and a word on hardware experiments**
>
> To address your question about Learning from Play, we have run experiments by giving LfP 2x and 4x the amount of playdata that LfP (and PLEX) used in the submission's experiments presented in the paper's Figure 2.
>
> The results on the 5 MW tasks are as follows:
>
> **LfP (original 0.5-noise playdata):**
>
> hand-insert: 22.0
>
> door-unlock: 54.0
>
> door-lock: 42.0
>
> bin-picking: 0
>
> box-close: 6.2
>
>
> **LfP (2x 0.5-noise playdata):**
>
> hand-insert: 24.4
>
> door-unlock: 59.0
>
> door-lock: 42.6
>
> bin-picking: 0.8
>
> box-close: 5.2
>
> **LfP (4x 0.5-noise playdata):**
>
> hand-insert: 25.0
>
> door-unlock: 62.2
>
> door-lock: 47.2
>
> bin-picking: 0
>
> box-close: 8.0
>
>
> These results show that LfP's performance doesn't change much with the *quantity* of play data. What LfP needs for better performance is *higher-quality* *visuomotor* playdata. This is exactly where PLEX's strength lies: PLEX can deal with low-quality playdata, which is much cheaper to collect, as long as it has access to good *video-only* data, which is available in quantity.
>
>
> We haven't been able to complete the long-horizon experiments you asked about, although we have started some on the new LIBERO benchmark (https://libero-project.github.io/intro.html), on its LIBERO-100 split, whose 10 evaluation tasks are longer horizon. However, we emphasize that we believe a more practical approach for using PLEX in long-horizon settings is in combination with LLM or VLMs that have universal task knowledge for high-level planning and breaking down a task execution into a sequence of skill applications. PLEX is meant to be the model that will plan out and execute each of the required skills.
>
> Regarding the hardware experiments, please see our latest response to reviewer **yTsm**.

---

### Official Review · Reviewer_yTsm · 2023-07-19

**Confidence:** 3
**Originality:** Good
**Technical Quality:** Good
**Clarity Of Presentation:** Good
**Impact:** 3

**Recommendation:**

Weak Reject: I recommend rejecting the paper, but will not argue for my recommendation if the majority of other reviewers have a different opinion.

**Review:**

Strengths:
* The PLEX architecture enables pretraining on different kinds of data, e.g. a large-scale video dataset and small-scale demonstrations.
* The experiments show strong performance for PLEX in zero-shot, finetuning, and training from scratch settings.

Weaknesses:
* The motivation for using multiple data sources (access to large-scale internet/human videos) does not apply well to most of the experiments. The experiments use pretraining data from the same simulation environment as the evaluation. PLEX+R3M does use internet data, but it is not being presented as the primary method. Further experiments using diverse, out-of-distribution pretraining data are needed to determine the benefits of this architecture.
* Some hyperparameters are different between MetaWorld and RoboSuite and between PLEX and DT. Is there motivation for the differences in number of layers (2 for PLEX Meta-World, 3 for PLEX Robosuite, and 4 for DT Robosuite)? The training steps per epoch is also different.

Minor suggestions:
* The introduction is quite long and overlapping with some sections. The second paragraph (L27-39) is very similar to section 2.2 and could be shortened/removed from the intro. L61-75 feels like it belongs in related work.
* Figure 1 is hard to follow. The font size on the diagram is too small to read easily. The caption is perhaps too detailed, in addition to its small font size and lack of margins.
* Typo: L286 samll -> small

**Quality Of The Limitations Section:**

Limitations are addressed clearly

**Questions For Rebuttal:**

* For any hyperparameters that differ across settings/baselines, how were the hyperparameters selected (e.g. detail the hyperparameter search)?

**Robotics Focus:**

Highly relevant to robotics but no hardware experiments

**Summary Of Paper:**

The paper introduces PLEX (planning-execution), a transformer architecture with a task-specific planner and a subsequent executor that outputs actions. Each component is trained on different data: the planner is pretrained on a larger video dataset without access to actions and outputs a plan in a learned observational embedding space. The executor is pretrained on a smaller visuomotor dataset with observations and actions. MetaWorld experiments show that PLEX can complete some tasks zero-shot (without finetuning on demonstrations). Robosuite experiments show that PLEX's relative positional encoding enables better performance than the DT baseline.

**Summary Of Recommendation:**

The idea of enabling multiple forms of pretraining/data is well-motivated. The experiments for the most part don't actually use different data sources (just the task/presence of actions is changed). Experiments with real-world pretraining data would strengthen the paper.

I'm not so familiar with the related work on transformers for manipulation, so I cannot determine the novelty/impact of this particular architecture. If most of the benefit is from relative positional encodings rather than the planner-executor split, the paper should be reframed.

---

> ### Author Response · Authors · 2023-08-16
> **Rebuttal cont'd -- on hyperparameter and hardware experiments**
>
> Please refer to our response to reviewer **aYEz** regarding some hyperparameter tuning experiments.
>
> Also, we would like to confirm that, although this submission doesn't have hardware experiments, it is our earnest intention to deploy PLEX on a real robot. In the past two weeks, we have taken steps towards this goal and trained several PLEX instances for a WidowX250 robot operating in a toy kitchen. Please refer to https://anonymous68546.github.io/plex/ for videos of our WidowX250 performing three tasks (*pick-and-place*, *lift-pan*, and *push-into-sink*) that we are currently teaching it.
>
> Specifically, our setup is related to the one used for collecting the Bridge Dataset (BD, https://rail-berkeley.github.io/bridgedata/), but differs from it in the positions of the cameras and the kitchen setup. Discrepancies in camera positioning (indeed, the Bridge Dataset doesn't contain info on the original camera poses) make our embodiment different from the embodiment in BD to the point that learning visuomotor dynamics for our setup using BD doesn't work. Therefore:
>
> -- We use BD as the *video* dataset $D_{mtvd}$, ignoring the action information in it.
>
> -- We collect the visuomotor dataset $D_{vmt}$ using our own embodiment. Currently it has ~300 trajectories of 160 steps each.
>
> -- We collect target-task demonstrations for the aforementioned tasks on our embodiment as well. Currently, we have 30 demonstrations per task.
>
> Our training pipeline still requires more work to determine PLEX hyperparameter values that yield policies robust to environment variations. However, already now, using a BD-pretrained PLEX with 3 executor layers and 3 planner layers as well as a ResNet-18 encoder, we are able to learn robot policies that are infeasible to learn from scratch using just 30 demonstrations per task in such a complex setup.

---

### Official Review · Reviewer_aM5x · 2023-07-24

**Confidence:** 4
**Originality:** Good
**Technical Quality:** Very Good
**Clarity Of Presentation:** Very Good
**Impact:** 3

**Recommendation:**

Strong Accept: I recommend accepting the paper and will argue for my recommendation even if other reviewers hold a different opinion.

**Review:**

Overall, I think the paper does a good job motivating the problem of learning from a lot of lower quality and little high quality data. This is a relevant problem for the robot learning community. The targeted data categories and learning system are well described as are the experiments.
I have some concerns about the design of datasets in the experiments and some of the explanations of the approach.

line 105: ‘many of these trajectories are generated by activities that most people will not find meaningful’ - In the presented experiments D_vmt and D_ttd are generated from the same agent and tasks with action noise added in D_vmt. It would be interesting to study the hypothesis where D_vmt is in fact a set of tasks that are not goal-oriented and ‘meaningless’ or at least the goals do not coincide precisely with D_ttd.

line 181 ‘When a modality is missing, it is replaced by trainable placeholder vectors during embedding’ - Is there an intuition as to why you use trainable placeholders? How does it compare, for example, to feeding randomised inputs or using dropout?

Equation (2) and surrounding text: I might be misreading this, but should the summation on the right-hand side start at t=T+1 instead of t=1+L ?

line 229: ‘[..], this involves deciding which part of PLEX to adapt’. - In the experiments, it seems that the last layers comprising 5% of the parameters are fine-tuned. Have you compared fine tuning to other different parts of the network or how did you ‘decide’ what to fine-tune in your system?

line 236: ‘behaviour cloning [..] is equivalent to a stopgrad-free version of L_EX’ - Could you elaborate on this? It seems L_EX is taking as input the past and a future state and regresses on an action, while BC usually takes in only past states and a task conditioning. Do you suggest the latter and remove the planner entirely?

line 254: ‘standard deviation of 0.5’ - What are the units here and how does 0.5 relate to the distribution of actions taken by the scripted policy, i.e. your D_ttd? This relation seems to be important for interpreting how close D_ttd is to D_vmt or how much D_vmt indeed represents play-like data without goals.

line 291: In the experiments D_vmt is a noisy version of D_ttd generated on the exact same tasks and with the same agent (the scripted policy). One explanation as to why adding actions does not help, is that PLEX might be able to connect image and action from D_vmt and image to goal from D_ttd. It might be an artefact of your choice of D_ttd and D_vmt. It would be interesting to conduct this study with a D_vmt that does not resemble the exact same tasks from D_ttd.

As noted in the limitations, the presented approach currently uses data from a single robot whereas the approach is to leverage large amounts of data that potentially stem from other robots. Different robots may have different proprioception and action representations. How would your approach extend from a single action / proprioception representation?


**Quality Of The Limitations Section:**

Limitations are addressed clearly

**Questions For Rebuttal:**

I think the weakest point of this paper is the fact that D_vmt and D_ttt were generated by the same policy and tasks. Could you discuss how choosing D_vmt with more ‘meaningless’ tasks would affect the study?

**Robotics Focus:**

Highly relevant to robotics but no hardware experiments

**Summary Of Paper:**

This paper proposes a framework for learning robotic tasks from multiple datasets of varying quantity and quality. An agent architecture is presented that consists of a planner mapping (embedded) observations onto desired future (embedded) observations and an executor that serves as an inverse model and maps observed and desired embedded states onto actions.
The transformer based architecture uses embeddings that are pre-trained on ImageNet and some of the high-quantity-low-quality datasets. Robotic data comprising actions is then used to further fine-tune the agent.
Experiments demonstrate effectiveness of the agent on a set of simulated benchmark tasks. Further, the paper demonstrates that relative position encodings appear to be superior to commonly used absolute position embeddings.


**Summary Of Recommendation:**

Overall, I recommend the paper for submission, because it is well motivated, the approach is explained clearly and the experiments examine the approach mostly well. My only concern is the choice of datasets in the experiments that might be overly simplistic. I would like to hear from the authors on this topic in the rebuttal.

---

### Author Response · Authors · 2023-08-11
**Common response**

First of all, we must say that we were **amazed** by the thorough and constructive reviews, even in cases when the reviewers weren’t sure their criticisms stood. It’s rare to see this level of review quality these days. Thank you.

**For now, in the interest of starting a discussion with the reviewers, we are posting:**
1. **A version of the paper revised per the reviews (an identical copy attached to each rebuttal).**
2. **Our responses to the individual reviews below, which refer to this revised version.**

The changes w.r.t. the original submission, both removals and additions, are highlighted in the pdf. If the removed sections are actually deleted, the paper will fit within the 8-page limit.

The most significant high-level change is a closer alignment between the paper’s claims and its contributions, per some of the reviewer comments. Namely, the paper is about an *architecture* for a foundation model (FM) rather than a FM per se. Training a full-blown FM based on PLEX will require an amount of in-the-wild videos and compute that we (the authors) don’t have at our disposal, and we don’t attempt it in this work. Rather, in this submission we illustrate the PLEX architecture’s properties that, due to PLEX’s data efficiency, manifest themselves even in smaller-scale experiments.

Among other things, we have toned down the claims about zero-shot (ZS) generalization. We *believe* that a FM based on PLEX will generalize ZS with high performance across a wide variety of tasks, but we don’t mean to claim that PLEX achieves this generalization breadth at the scale of this paper’s experiments – in fact, this paper mainly focuses on PLEX’s post-finetuning performance. What we *do* mean to demonstrate is that *even* at the moderate pretraining scale in this paper, PLEX already exhibits non-trivial ZS performance. Hopefully, the revised submission reflects this more accurately.

**In the meantime, we are also trying to run additional experiments in response to some of the reviewers' questions/requests, and we hope to post:**

3. **The additional experiment results by August 15.**

---

### Decision · Program_Chairs · 2023-08-30

**Decision:**

Accept (Poster)

**Comment:**

The authors propose an architecture for learning robot policies from a mix of three types of data sources: (i) video-only data, (ii) non-task-specific sensorimotor data, and (iii) task-specific sensorimotor data.  The architecture has a few components which are trained in a prescribed fashion: a latent-space planner which can be trained from (i), and a goal-conditioned / inverse-model they call an executor.  Experiments show they are able to use this combination of different data sources with some success, for example in a way they outperform the "Learning from Play" baseline which is not designed to use (i).  While there are many other prior works that can also use (i), the work is perhaps relatively unique in its combination of all (i), (ii), (iii).

The reviewers had a variety of opinions.  Common strengths pointed to include that it is nice the method can do training on these different types of data, has strong performance in some settings, and provides useful empirical analysis.  The weaknesses mentioned mostly center around what I would characterize as an opinion by the reviewers that the authors have overclaimed their results, and/or that the results don't match the claims.  As reviewer yTsm puts it (paraphrased) "the motivation does not well apply to most of the experiments, pretraining from the same env as the eval."  Reviewer aYEz also had several critiques on the claims using the term "zero-shot", which were in part addressed in the rebuttal, but remains in their final determination still offput by the inaccuracy of the "in-the-wild" claim.  Although the other 2 reviewers suggested Strong Accept, these two reviewers (yTsm, aYEz) stood by their Weak Reject suggestions, with valid and well-argued concerns, despite the rebuttal attempts by the authors.

After careful consideration I would like to recommend the following:

- First, the reviewers should see the authors' revised manuscript which removes any mention of the term "in-the-wild" (https://openreview.net/attachment?id=iJx-Ksnsz4&name=file_upload -- I think that hyperlink will still work).  I think this partially relieves the dissenting opinion of aYEz in particular, and in part yTsm.
- However, even though the revised manuscript is a step in a good direction, the authors should amend their statements *further*... specifically, although there is no "in-the-wild" term used, the authors cite "[4-8]", which are in-the-wild video datasets.  This implies that their method will use these types of data, particularly since paragraph 2 says, "PLEX achieves exactly that".
- The authors should amend the introduction so that it does not imply this.  A suggested workaround is rather than to characterize (1), (2), (3) different types of "datasets" in the introduction paragraph, the authors can characterize three different "classes of data", and simply state that it is interesting to study methods that can use video-only data, even if the dataset comprised of such data is taken from the identical environment from which the robot actions could have simply been used -- which is the case in this paper's experiments.  The authors should only refer, even implicitly through citation, to in-the-wild video datasets in the context of related or future work.
- This is important because words matter!  There are many, for example, PhD students trying to work towards the vision of actually using in-the-wild video data in a way that is broadly useful for robotics.  This paper does not accomplish that goal, and should not imply that it does.  It can be viewed as perhaps a step towards that goal, but it does not arrive there.
- Similarly, also important, the authors should remove the notion of "embodiments" plural from the Section B, Problem formalization, because there are no experiments that address the multi-embodiment case.  This is classic over-generalization in a problem formalization.
- Regardless, if we re-examine both reviewers yTsm's and aYEz's dissenting opinions with the writing modifications requested as above, I think they would feel much more comfortable with accepting the paper.  The results are still interesting as they are truthfully, and don't need to be oversold.